# Peptide Receptor Radionuclide Therapy (PRRT) with ^177^Lu-DOTATATE; Differences in Tumor Dosimetry, Vascularity and Lesion Metrics in Pancreatic and Small Intestinal Neuroendocrine Neoplasms

**DOI:** 10.3390/cancers13050962

**Published:** 2021-02-25

**Authors:** Ulrika Jahn, Ezgi Ilan, Mattias Sandström, Mark Lubberink, Ulrike Garske-Román, Anders Sundin

**Affiliations:** 1Department of Surgical Sciences, Radiology & Molecular Imaging, Uppsala University Hospital, SE-751 85 Uppsala, Sweden; Ezgi.ilan@akademiska.se (E.I.); mattias.sandstrom@akademiska.se (M.S.); mark.lubberink@akademiska.se (M.L.); ulrike_garske@hotmail.com (U.G.-R.); anders.sundin@radiol.uu.se (A.S.); 2Medical Physics and Institute for Immunology, Uppsala University Hospital, SE-751 85 Uppsala, Sweden; 3Medical Imaging Centre, Uppsala University Hospital, SE-751 85 Uppsala, Sweden; 4Department of Immunology, Genetics and Pathology, Rudbeck Laboratory Uppsala University, SE-751 85 Uppsala, Sweden

**Keywords:** radionuclide therapy, theranostics, ^177^Lu-DOTATATE, PRRT, small intestinal NEN, pancreatic NEN, dosimetry, low-dose-radiation, vascularization

## Abstract

**Simple Summary:**

Patients suffering from disseminated, progressive, neuroendocrine neoplasms with a sufficient amount of somatostatin receptors and good kidney function can be treated with radioactive hormone-like molecules to prolong their life. In this study, the radioactivity in one tumor per patient at each treatment cycle was calculated and compared between 23 patients with pancreatic and 25 patients with small intestinal neuroendocrine neoplasia. Both types of tumors absorb a larger amount of radioactivity during early cycles that subsequently decline in the later cycles. This finding was more pronounced in the pancreatic tumors, which also expressed higher blood perfusion in the early cycles, known to facilitate the effect of radiation. This could be part of the reason why the pancreatic tumors shrunk more rapidly than the small intestinal ones. Our results also imply that increased administered activity in the early therapy cycles may be beneficial, at least in pancreatic neuroendocrine tumor patients.

**Abstract:**

Dosimetry during peptide receptor radionuclide therapy (PRRT) has mainly focused on normal organs and less on the tumors. The absorbed dose in one target tumor per patient and several response related factors were assessed in 23 pancreatic neuroendocrine neoplasms (P-NENs) and 25 small-intestinal NEN (SI-NENs) during PRRT with ^177^Lu-DOTATATE. The total administered activity per patient was (mean ± standard error of mean (SEM) 31.8 ± 1.9 GBq for P-NENs and 36 ± 1.94 GBq for SI-NENs. The absorbed tumor dose was 143.5 ± 2 Gy in P-NENs, 168.2 ± 2 Gy in SI-NENs. For both NEN types, a dose–response relationship was found between the absorbed dose and tumor shrinkage, which was more pronounced in P-NENs. A significant drop in the absorbed dose per cycle was shown during the course of PRRT. Tumor vascularization was higher in P-NENs than in SI-NENs at baseline but equal post-PRRT. The time to progression (RECIST 1.1) was similar for patients with P-NEN (mean ± SEM 30 ± 1 months) and SI-NEN (33 ± 1 months). In conclusion, a dose response relationship was established for both P-NENs and SI-NENs and a significant drop in the absorbed dose per cycle was shown during the course of PRRT, which warrants further investigation to understand the factors impacting PRRT to improve personalized treatment protocol design.

## 1. Introduction

In the prospective randomized controlled study (NETTER-1) in patients suffering from inoperable, disseminated, progressive small intestinal neuroendocrine neoplasms (SI-NENs), peptide receptor radionuclide therapy (PRRT) with ^177^Lu-DOTATATE was proven superior to high-dose somatostatin analog (SSA) therapy alone, resulting in longer mean progression-free survival (PFS) [1]. PRRT in patients with other NEN types have also shown favorable outcomes in long-term follow-up studies, most of which, however, are retrospective and non-randomized [2,3,4]. NENs are, as an entity, markedly heterogenous as a result of their differences in primary tumor sites, their degree of proliferation and functional status [5,6]. As a consequence, they may respond differently to PRRT, which, for example, can be demonstrated by the varying response patterns between pancreatic NENs (P-NENs) and SI-NENs [7]. P-NEN patients generally demonstrate apparent morphological fractional (i.e., the percentage change between baseline and best response, ∆%) tumor shrinkage and are thus more often classified as complete or partial responders (CR, PR) [8,9,10]. In SI-NEN patients, the tumors, by contrast, generally remain unchanged in size during PRRT and they are more often classified as having a stable disease (SD) [7,9,11]. The Rotterdam standard PRRT protocol, consisting of four cycles with 7.4 GBq of ^177^Lu-DOTATATE [12], has been applied in the majority of previous studies, and was also used in the NETTER-1 trial. When PRRT was introduced in our center in 2005, it was required by the regulatory authorities that dosimetry should be performed at each cycle. Thus, due to this demand, it became possible to tailor the number of PRRT cycles individually for each patient and, in parallel to dose planning in external radiation therapy, a dosimetry-guided PRRT protocol was established. The absorbed dose in the dose-limiting organs was used to define the maximum number of PRRT cycles for each patient, applying an upper limit of 23 Gy and 2 Gy to the kidneys and bone marrow, respectively [13].

Dosimetry at PRRT in most reports is focused on the normal, most radiation-sensitive organs, such as the kidneys and bone marrow [8,14]. Accumulation of ^177^Lu-DOTATATE in the tumor tissue has been much less explored, and when mentioned, data are mostly presented combined [8] or as tumor-to-normal-tissue ratios (%), usually to the liver and/or kidneys [4,9,15]. 

In previously published reports on tumor dosimetry, in 24 P-NEN and 25 SI-NEN patients, respectively, a relationship was found between the absorbed dose and the maximal fractional tumor shrinkage in P-NENs [16] but not in SI-NENs for which, instead, a correlation was found between the administered activity (GBq) and tumor shrinkage [17].

The aim of the present study was to evaluate and compare several tumor-related and patient-associated parameters in order to better understand the mechanisms involved in the different responses to PRRT between these two NEN types. The considerably longer follow-up time of the present study, as compared to that in the previous publications, additionally allowed for a more accurate assessment of the PRRT outcome. 

## 2. Result

### 2.1. Patients

Patient and tumor characteristics at baseline are shown in Table 1.

The follow-up time was a median of 42 (range 7–110; mean ± SEM 49 ± 4) months. During this time, only four SI-NET patients with G1 tumors (Ki-67 ≤ 2%) were without disease progression according to RECIST 1.1. Median PFS was similar for P-NETs and SI-NETs (*p* > 0.05) Table 2.

### 2.2. Tumors

The tumor volumes were measured on baseline CT by using a semi-automated software and were additionally calculated from the measurement of the transverse tumor diameter (4πr^3^/3), thus assuming a spherical tumor shape. 

At baseline, the transverse tumor diameters of the selected lesions were similar for P-NENs (mean ± SEM) 58 ± 6 mm and SI-NENs 44 ± 4 mm (*p* > 0.05). The volumes calculated by the semi-automated method were significantly smaller than when calculated from the transverse tumor diameters (P-NENs *p* = 0.01, SI-NENs *p* = 0.03).

The semi-automated volumes at baseline for P-NENs (mean ± SEM) 72 ± 18 cm^3^ and SI-NENs 44 ± 19 cm^3^ were similar when all patients were included (*p* > 0.05). When one extreme outlier in the SI-NEN group (10 times the mean volume) was removed, the difference in semi-automated volumes became significant (72 ± 18 cm^3^ and 26 ± 5 cm^3^, respectively) (*p* = 0.02). As for the RECIST 1.1 measurements at baseline, P-NENs (mean ± SEM) 117 ± 15 mm and SI-NENs 114 ± 9 mm were similar (*p* > 0.05).

From baseline to the best response, the fractional decrease of the tumor diameters, the semi-automated volumes and RECIST 1.1 values for the P-NENs exceeded that of the SI-NENs, as shown in Table 3. 

These findings still held true even when exclusively considering the G2 (Ki-67 3–20%) patients, 78% P-NENs (18/23) and 40% SI-NENs (10/25).

The mean time from the start of PRRT to the best response was similar (*p* > 0.05) but tended to be longer for SI-NENs than for P-NENs. The maximal fractional tumor shrinkage regarding the diameter and semi-automated volume in P-NENs occurred after (mean ± SEM) 19 ± 3 months and 19 ± 3 months, respectively, and in SI-NENs after 25 ± 4 and 26 ± 4 months, respectively.

At baseline CT, the tumor attenuation (Hounsfield units, HU) in the pre-contrast (native) phase was higher in SI-NENs (mean ± SEM) 42 ± 1 HU than in P-NENs 37 ± 1 HU (*p* < 0.001). Nonetheless, the tumor heterogeneity, assessed as the standard deviation of the native tumor attenuation measurements, was similar in SI-NENs and P-NENs (mean ± SEM) 12 ± 1 HU and 14 ± 1 HU respectively (*p* > 0.05). At baseline, 23 P-NEN patients and 22 SI-NEN patients underwent contrast-enhanced CT examinations; thus, three SI-NEN patients did not receive intravenous contrast due to their medical status. The degree of vascularization at baseline, as reflected by tumor-to-aorta HU ratio in the late arterial contrast-enhancement phase, was significantly higher in the P-NENs (mean ± SEM) 49 ± 4 than in the SI-NENs 30 ± 3 (*p* < 0.001). 

At the time of best response, as regards the semi-automated volume, after three to nine months, only 21 P-NEN patients and 20 SI-NEN patients underwent contrast-enhanced CT. In the majority of lesions, the tumor-to-aorta attenuation (HU) ratio decreased significantly from baseline in the P-NEN group (*p* < 0.01) but not in the SI-NEN group (*p* > 0.05). For some of the NEN lesions, 7/21 (33%) P-NENs and 5/20 (25%) SI-NENs, there was instead an increase of the tumor-to-aorta attenuation (HU) until best response, as shown in Figure 1. After the time of best response, the vascularization ratio of the P-NENs leveled out to that of the SI-NENs and then remained stable over the course of follow-up.

### 2.3. Administered Activity (GBq)

The total amount of administered activity was 731.5 GBq to the P-NEN group and 899.3 GBq to the SI-NEN group. Due to the clinical and hematological status of the patients, the administered activity for some of the PRRT cycles was reduced. In the P-NEN group, this was the case in 9/100 (9%) cycles in 5/23 (22%) patients, resulting in a median of 29.6 GBq (range 13.4–44.4; mean ± SEM 31.8 ± 1.9 GBq) per patient. For the same clinical reasons, the time interval between treatment cycles was extended beyond eight weeks in 4/100 (4%) cycles in 4/23 (17%) P-NEN patients. Correspondingly, in the SI-NEN group, the administered activity was reduced for 10/124 (8%) cycles in 5/25 (20%) patients with a subsequent median of 37 GBq (range 20.8–51.8; mean ± SEM 36 ± 1.9 GBq) per patient [17]. The time interval between treatment cycles was extended for 27/124 (17%) cycles in 15/25 (60%) of the SI-NEN patients. Thus, the total amount of administered activity throughout the course of PRRT was similar in both groups of NEN patients (*p* > 0.05), as was the accumulated administered activity until best response, i.e., the maximum fractional shrinkage of the tumor diameter, semi-automated volume and RECIST 1.1 results. In the SI-NEN group, the correlation between accumulated administered activity and RECIST 1.1 values (*p* = 0.003) was consistent with our earlier report [17], but no such correlation was found in the P-NENs, as shown in Figure 2.

The previously reported [17] correlation between the accumulated administered activity and the maximum fractional shrinkage of the semi-automated volume in the SI-NEN group could not be corroborated in the present study with a considerably longer follow-up time. There was no correlation between the administered activity (GBq) and the maximum fractional shrinkage of tumor diameter or semi-automated volume, in any of the groups. In the P-NEN group, the total administered activity correlated to the progression-free time from the start of PRRT (*p* = 0.03), but there was no such correlation in the SI-NEN group.

### 2.4. Absorbed Dose (Gy)

The total absorbed dose in the selected tumors for all cycles was similar (*p* > 0.05) for P and SI-NEN, with (mean ± SEM) 143.5 ± 2 Gy in P-NENs and 168.2 ± 2 Gy in SI-NENs. The extended seven-day dosimetry protocol was applied at the first cycle and in 20 P-NEN patients and 19 SI-NEN patients was performed at cycle 3 or 4, corresponding to four to eight months from baseline. Between baseline and the second seven-day-protocol dosimetry (cycle 3/4), there was a significant decline in the the median absorbed doses in the P-NENs (*p* < 0.01) but not in the SI-NENs (*p* > 0.05), as shown in Figure 3.

Likewise, a decline was seen in the ratio between the absorbed dose and administered activity (Gy/GBq), shown in Table 4. 

In general, when considering all dosimetry calculations, including those with a short 24 h dosimetry protocol, the median absorbed dose declined from the first to the last cycle and was more pronounced in P-NENs than in SI-NENs, as shown in Figure 4.

Notably, there were some lesions that deviated from this general pattern, with increasing absorbed doses during one or multiple cycles, as shown in Figure 5.

The accumulated absorbed dose until best response as regards tumor diameters, semi-automated volumes and RECIST 1.1 values was similar in the two NEN types (*p* > 0.05), as shown in Table 5.

The present study corroborated the previously reported dose–response relationship between the absorbed dose to the tumor and the fractional shrinkage of the lesion diameter in P-NENs [16]. As for the SI-NENs, with this longer follow-up, a dose–response relationship was found given that one extreme outlier, with a large uptake and minute shrinkage, which was removed (*p* = 0.001)—see Figure 6.

The correlation between the accumulated absorbed dose and maximal fractional semi-automated volume shrinkage was significant and rather similar between P-NENs and SI-NENs, as Figure 7 shows.

When calculating the accumulated absorbed dose to semi-automated volume there was no apparent extreme value in the SI-NEN group. 

The P-NEN patients accordingly demonstrated a dose–response relationship between the absorbed dose to tumor and best RECIST 1.1 values (*p* = 0.01) but a comparable correlation was lacking in the SI-NEN group. 

## 3. Discussion

This is, to our knowledge, the first comparative study on absorbed tumor doses in P-NENs and SI-NENs, with a focus on the individual cycles during PRRT. In previous publications, our group reported a dose–response relationship in 24 P-NENs receiving PRRT with ^177^Lu-DOTATATE [16] but found no such correlation for 25 SI-NENs (even though a correlation between the administered activity and morphologic response was found for the latter group) [17].

In the present study, we compared the absorbed doses in individual tumors, cycle by cycle, in 23 and 25 of the earlier-reported P-NENs and SI-NENs respectively, and the results after a considerably longer follow-up (median 42 months). The aim was to assess the similarities and differences between the two NEN types regarding administered activity, absorbed doses and the patterns of change during the course of PRRT, including the tumor vascularity and tumor shrinkage in response to treatment. The study resulted in two major findings; firstly, that the vascularization in most P-NENs was high at baseline CT but decreased at subsequent CT monitoring during the course of PRRT (Figure 1), and secondly, that the median absorbed dose in both tumor types dropped at consecutive PRRT cycles and significantly between the first and last complete seven-day dosimetry cycle in P-NEN but not in SI-NEN (Figure 3).

Our observations of several deviating factors between P-NENs and SI-NENs support what has been reported in previous PRRT studies on the differences between these two NEN types [7,9,10,11], but are nevertheless difficult to explain. The CT measurements of tumor contrast-enhancement strongly indicate a reduction in tumor vascularization (Figure 1), which may have bearing on our findings. The vascularity of primary pancreatic tumors and P-NENs has earlier been analyzed and was found to correlate to tumor classification and survival [18,19]. Also, fold changes of contrast-enhancement in P-NEN metastases has been shown to correlate to PRRT outcome [20]. These findings are in agreement with preclinical studies on vascularization in radiated tumors in mice [21].

The following discussion relates to factors that, at least partly, may be more speculative, owing to the fact that tumor dosimetry in the framework of PRRT so far is little explored.

The initial high vascularization in the P-NENs, but not in the SI-NENs, during the first PRRT cycles may have implications for the delivery of the ^177^Lu-DOTATATE preparation into the tumors, as suggested by the generally higher absorbed dose found early in the P-NEN lesions and the subsequent decline over the course of PRRT (see Figure 3 and Figure 4). With their higher tumor vessel densities, not only was the radioactivity delivery and distribution throughout the P-NEN volume likely to be facilitated, but their higher degree of vascularization might also have increased tumor oxygenation, and hence facilitated the effects of the delivered radiation. After completion of PRRT, the tumor vascularization decreased in the majority of P-NET lesions, as reflected by their CT contrast-enhancement. A corresponding reduction in P-NEN vascularization between the CT at baseline and on follow-up CT post-PRRT has previously been reported by Pettersson et al. [20]. Thus, although the total administered activity (GBq) was similar in P-NEN and SI-NEN patients, the decline in absorbed doses (Gy) in the tumors over consecutive PRRT cycles was significant in the majority of P-NEN patients, as shown in Figure 3 and Figure 4. This was especially apparent between the first two PRRT cycles (see Figure 4 and Figure 5) and constitutes a finding that, to the best of our knowledge, has not previously been described.

Individual tumors do not only consist of the malignant cell clone itself, but comprise a complex mixture of tumor vessels, supportive tissues, tumor fibroblasts and immunoactive cells, all with varying likelihoods of responding to therapy [22]. This needs to be considered as a contributing factor when trying to understand the current different response patterns between the P-NENs and SI-NENs. It is also well-understood that the effect of low-dose-rate radiation is difficult to evaluate and that biological effects depend on both the cellular and intracellular distribution of the radionuclide. Thus, even if the macroscopic dose to the tumor is similar, the effects can be different on a cellular and subcellular level [23,24]. This was illustrated by the differences in tumor shrinkage between P-NENs, and SI-NENs despite similar administered activity and absorbed dose.

The CT attenuation values in the native phase at baseline were lower in P-NENs than in SI-NENs. Since the standard deviations of these attenuation measurements reflecting tissue heterogeneity were the same for both NEN types, varying the degree of tumor necrosis is unlikely to explain this attenuation difference. The occurrence of fibrosis in NENs is an area that is not yet fully understood [25,26]. Hence, the higher attenuation in the SI-NENs may support the notion that these tumors comprise more fibrosis than P-NENs, which may be contemplated as a contributing factor in the different metric responses to PRRT between the two NEN types.

In order to achieve a conventional evaluation on the patients’ tumor disease, the PRRT response was also assessed according to RECIST 1.1 [27]. We could corroborate the findings from the earlier study in SI-NEN patients of a correlation between administered activity (GBq) and RECIST 1.1 measurements, and with a now slightly higher explanation value of 33% (compared to 28%). In the P-NEN patients, however, we failed to find a correlation in this regard (Figure 2). Despite all the presented differences between the NET-types, the median of PFS was nevertheless similar for P-NEN and SI-NEN patients (Table 2).

Since NEN lesions usually do not grow in a spherical fashion, and the transversal diameter therefore may not optimally reflect the tumor volume, we additionally assessed the relation between the absorbed dose and semi-automated volume measurements and found this correlation more consistent between the two NEN types than when using the tumor diameter (Figure 6 and Figure 7). Additionally, in the direct comparison of methods, volumes calculated from measurements based on the transversal tumor diameter were, for both NEN types, significantly larger than the corresponding semi-automated volumes. Notably, the volumes based on the transversal tumor diameters also revealed a much larger variation than the more precise semi-automated ones. Although more time-consuming, the semi-automated method is probably more accurate when assessing and comparing the treatment response in these two NEN types.

We are aware of the limitations of this study, being both retrospective, small and non-randomized. Because of the inherent limitations of the available dosimetry method, only one tumor per patient was analyzed in this study. The various metrics applied on this lesion may not necessarily have been representative for the whole tumor burden of the respective patient at any time point of the measured PRRT parameters. Similarly, related to the technical requirements of the dosimetry method, there was an inevitable bias in the patient selection. Consequently, merely a small fraction of patients and tumors out of the original PRRT study cohort was eligible for inclusion in the present evaluation, making the statistical power limited.

## 4. Materials and Methods

### 4.1. Patients and Tumor Selection

The patients were retrospectively selected from the previously published dosimetry-guided, prospective ^177^Lu-DOTATATE study (EudraCT 2009-012260-14) [7] and these subjects were also included in the two earlier reports on tumor dosimetry [16,17]. The study was approved by the Regional Ethical Committee and the Radiation Ethics Committee (2009/320; 2010/177). All of the patients provided written informed consent and all procedures were performed in accordance with the 1964 Helsinki declaration and its later amendments and comparable ethical standards.

All patients suffered from progressive, non-operable, disseminated NEN disease with high tumor uptake, Krenning grade 3–4, on somatostatin receptor scintigraphy with ^111^In-DTPA-D-Phe^1^-octreotide [28]. Out of 108 SI-NEN patients and 48 P-NEN patients in the EudraCT 2009-012260-14 study, complete dosimetry was achieavable for one single tumor in each of the 25 SI-NEN patients previously detailed [17] and 24 P-NEN patients. The principal reasons to exclude tumors from the dosimetric calculations were conglomerate growth (not allowing for separate lesion measurents), size ≤ 2.2 cm (because of partial volume effects) poor visibility on CT (unreliable delineation precluding size measurements) and if the tumor uptake did not follow the assumption of a single exponential decay, as required to allow for the dosimetric calculations [16]. In the present study, allowing for a longer follow-up time, one of the patients in the earlier report on P-NENs had to be excluded since the tumor during the later stages of PRRT shrunk below 2.2 cm in diameter and, thus, did not qualify for dosimetric calculation. The final cohort comprised 25 SI-NEN patients—16 men and 9 women—median age 64 years (range 18–76) and 23 P-NEN patients—12 men and 11 women—median age 59 years (range 40–76) all with histopathologically confirmed NENs. In the 25 SI-NEN patients, the analysed tumors all were metastases (21 liver and four lymph nodes) wheras in the 23 P-NEN patients, four primary tumors and 19 liver metastases were analysed. The patients’ liver tumor load was assessed on scintigraphy during the first PRRT cycle utilizing an arbitrary scale where <10% of the liver volume representing metastases was denoted I, 10–50% metastases II, >50% metastases III and >50% metastases & hepatomegaly IV. Patient and tumor characteristics are shown in Table 1.

### 4.2. ^177^Lu-DOTATATE Therapy

PRRT was administered acoording to previously published procedures applying a dosimetry-tailored treatment protocol [13,29,30]. Each PRRT cycle comprised 7.4 GBq of ^177^Lu-DOTATATE in 100 mL of saline that was infused intravenously for 30 min. For kidney protection, a mixed amino acid solution was infused intravenously. The P-NEN patients received between two and six PRRT cycles and the SI-NEN patients received between three and seven cycles. During the observation time of the present study, the patients received somatostatin analogs (SSA) except for PRRT.

### 4.3. Dosimetry

Dosimetry of the selected tumors was performed as previously detailed [16]. Briefly, a complete dosimetric evaluation, a seven-day protocol was performed at the first PRRT cycle and then at every third or fourth cycle (at four to eight months from baseline). The dosimetric calculations were based on SPECT/CT acquisitions at day one, four and seven post-^177^Lu-DOTATATE administration. A limited dosimetry protocol was used for the intermediary cycles, based on SPECT/CT acquired 24 h post ^177^Lu-DOTATATE administration, with the assumption of a similar effective half-life as for the previous complete dosimetry [30]. Hence, the absorbed dose to all the selected tumors was calculated at each PRRT cycle. A HERMES workstation (Hybrid PRD version 1.4 B, HERMES Medical Solutions AB, Stockholm, Sweden) was used to delineate the tumors by automatic threshold volumes of interest (VOIs) applying a 42% iso-contour, and corrected for partial volume effects, based on results from previous phantom measurements [16]. As the method stipulates that the entire selected tumor constitutes one VOI, it needs to be clearly separated from adjacent lesions in order to avoid overlap. To allow for at least 50% recovery of the activity concentration in the SPECT measurements, the tumor must measure ≥ 2.2 cm in diameter during the whole course of PRRT [16]. 

### 4.4. Clinical Symptoms and Adverse Events 

The occurrence of adverse events during PRRT and follow-up data were extracted from the patients´ digital records. SI-NEN related symptoms, such as both flushing and diarrhea, were experienced by the majority of SI-NEN patients (15/25, 60%). One of either symptom was recorded by 9/25 (36%) patients and one patient (4%) was free of symptoms [17]. In the P-NEN group, no patient reported hormonal symptoms, although seven of them had biochemical evidence of raised hormonal production (pancreatic polypeptide in four patients and calcitonin, glucagon and gastrin each in three patients).

### 4.5. Tumor Response Measurements

Intravenously contrast-enhanced CT/MRI was performed according to clinical routine examination protocols for NEN imaging with a 3 mm CT slice thicknesses and 5 mm MRI sections. Radiological follow-up utilized CT in all patients, except one SI-NEN patient who exclusively underwent MRI. CT/MRI was performed at baseline within three months before the start of PRRT, and before cycle 3 and 5. Following PRRT, CT/MRI was scheduled at 3, 6 and 12 months after the last cycle and then yearly. Because of logistical reasons, especially concerning our foreign patients, some variations in the follow-up protocol were encountered. By this longer follow-up we were able to receive and evaluate complete tumor response information on CT/MRI, including time to progression for the majority of patients and to calculate PFS. All but four SI-NEN patients progressed during follow-up. All examinations were re-assessed by an experienced radiologist (UJ, with 15 years of practice).

For each selected tumor, the maximum transversal diameter was measured and the volume calculated as previously described [17]. Briefly, semi-automated volume measurements utilized the CT/MR software supplied by the vendor (Carestream Vue PACS, Lesion Management, Version 12.0.0.7). In the only patient who was monitored by MRI, the transversal T2-weighted images were used for semi-automated volume measurements. For matters of comparison, the tumor volume was also calculated based on the transverse CT diameter (4πr^3^/3), assuming a spherical tumor shape.

Best response from baseline, regarding the selected tumor diameters and volumes, was calculated as the fractional and the absolute (mm and cm^3^) change. The general tumor response of each patient was also assessed according to RECIST 1.1 [27].

The degree of heterogeneity in the selected tumors was assessed on the pre-contrast (native) phase CT at baseline, applying 10 mm slices, in which the mean tumor attenuation (HU) and the standard deviation (SD) of the tumor attenuation was measured. As an appraisal of the tumor vascularisation, the mean tumor attenuation in the late arterial contrast-enhancement phase was measured in 10-mm-thick slices, on baseline CT/MRI, the first examination after start of PRRT and at the time of best response regarding tumor semi-automated volume. In order to correct for differences regarding the amounts of contrast medium and injection rate between examinations, the mean tumor attenuation was divided by the mean attenuation of the abdominal aorta, at the level of the celiac trunc, to form the tumor-to-aorta ratio.

### 4.6. Statistical Methods

PFS was measured from the first PRRT cycle to radiological progression according to RECIST 1.1 or death. PFS was calculated with Kaplan–Meier analysis. The box plots, linear regressions, Kaplan–Meier analysis and Wilcoxon signed-rank test utilized the JMP 12.0.1 software package (SAS Institute Inc., SAS Campus Drive, Cary, North Carolina 27513, USA). Statistical significance at linear regression was considered at a *p* < 0.05 with an explanation value of r^2^ ≥ 25%. The descriptive statistics and paired *t*-tests were calculated in Microsoft Excel and the differences within groups were assumed to be significant at 95% level using Independent t-test and manually calculated confidence intervals; GraphPad Prism, Version 6.07 was used for illustrating correlations and confidence intervals.

## 5. Conclusions

The absorbed dose (Gy) per cycle and the morphologic response differed between P-NEN and SI-NEN lesions, despite similar administered activity (GBq). A significant correlation between the accumulated absorbed dose (Gy) and tumor lesion shrinkage was found for both NEN types, although no correlation between the delivered accumulated activity (GBq) and morphologic response in the tumor lesions could be established. P-NENs on baseline CT had lower attenuation but were better vascularized than SI NENs, as reflected by the tumor-to-aorta ratio, and showed a steeper decline during therapy in this regard.

These findings warrant further investigation to better understand the various factors impacting PRRT, in order to improve and personalize the treatment-protocol design.

## Figures and Tables

**Figure 1 cancers-13-00962-f001:**
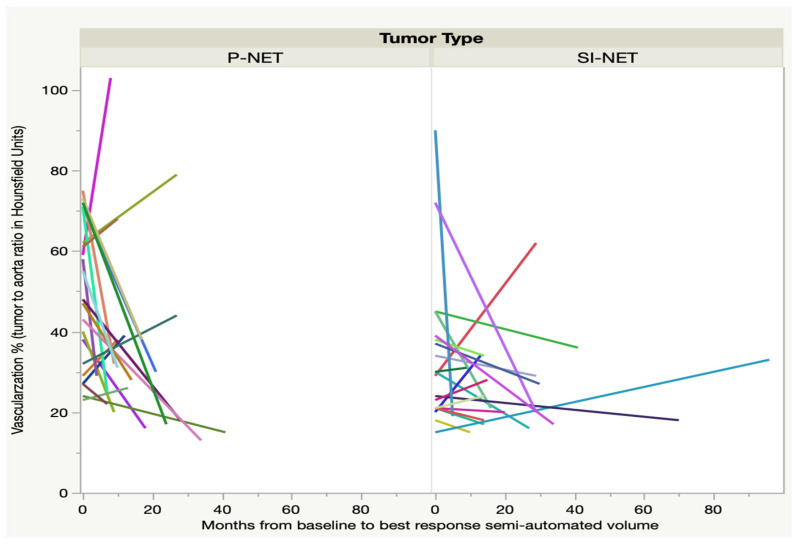
Fractional change (∆%) of the tumor to aorta ratio (as an indication of tumor vascularization) from CT at baseline to best response, as regards fractional semi-automated volume. Each line represents one tumor.

**Figure 2 cancers-13-00962-f002:**
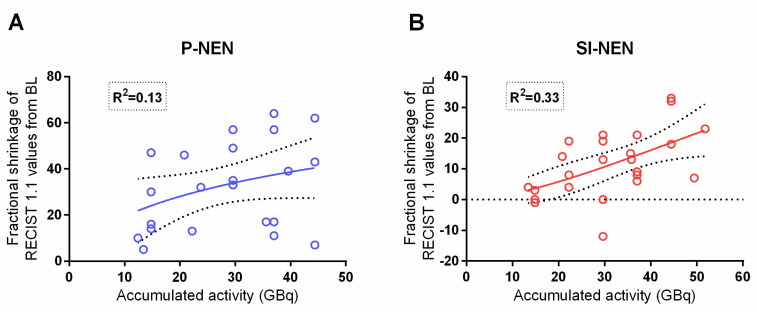
Correlation between accumulated activity (GBq) and RECIST 1.1 values. Accumulated activity (GBq) versus RECIST 1.1 values. (**A**) P-NENs (*n* = 23) and (**B**) SI-NENs (*n* = 25). P-NENs (*p* = 0.1) (**B**) SI-NENs (*p* < 0.01). Two patients in each group have identical values. BL; baseline.

**Figure 3 cancers-13-00962-f003:**
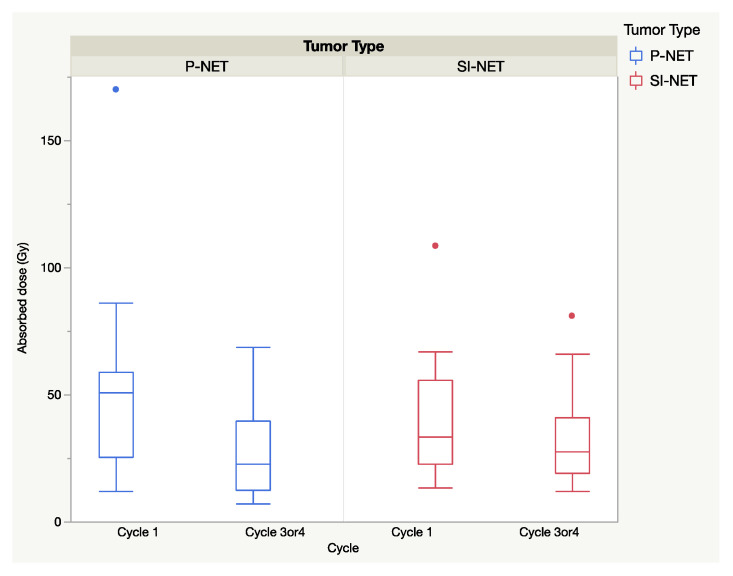
Absorbed dose (Gy) to the tumor at the first and the second cycle of full seven-day dosimetry (cycle 1 and 3/4) P-NENs (*n* = 20) and SI-NENs (*n* = 19).

**Figure 4 cancers-13-00962-f004:**
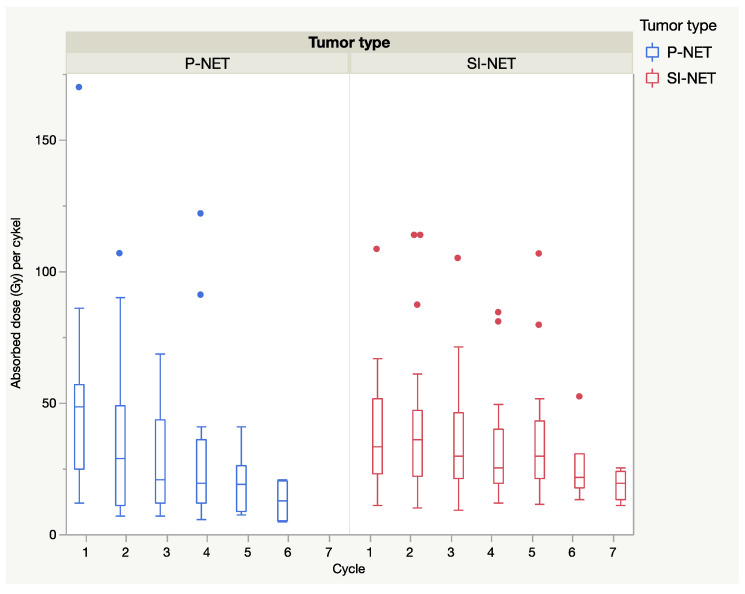
Tumor dose at each PRRT cycle. Median absorbed dose (Gy) in the measured tumors at each cycle. Full seven-day dosimetry cycles are shown together with 24 h dosimetry results. The absorbed tumor doses showed a continual decrease during the course of PRRT in both P-NENs (*n* = 23) and SI-NENs (*n* = 25).

**Figure 5 cancers-13-00962-f005:**
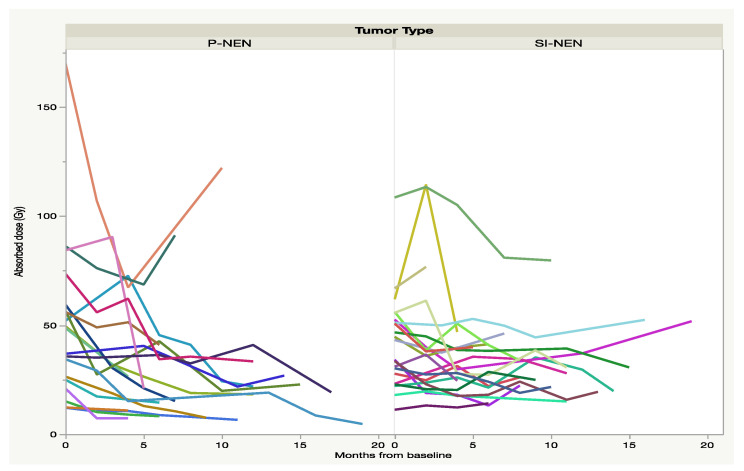
P-NEN tumor dose (Gy) at each cycle (*n* = 20) and SI-NEN tumor dose (Gy) at each cycle (*n* = 21) in patients with low to moderate tumor burden. Liver tumor burden I & II was assessed on scintigraphy during the first PRRT cycle (Table 1). Each line represents one tumor.

**Figure 6 cancers-13-00962-f006:**
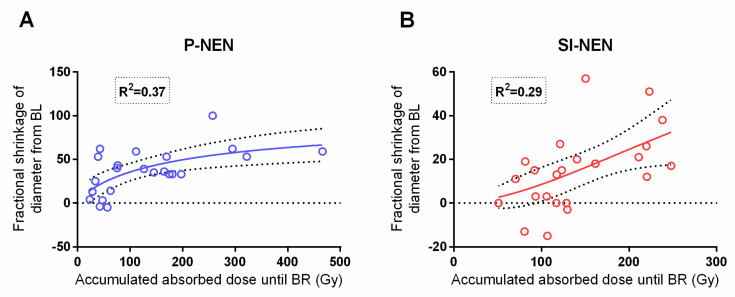
Comparison between tumor dose (Gy) and diameter shrinkage in P-NENs versus SI-NENs. Accumulated absorbed dose (Gy) to the tumor versus maximum fractional diameter shrinkage in P-NEN lesions (**A**) (*n* = 23) and SI-NEN lesions (**B**) (*n* = 24). P-NEN (*p* = 0.004), SI-NEN (*p* = 0.001). BL; baseline.

**Figure 7 cancers-13-00962-f007:**
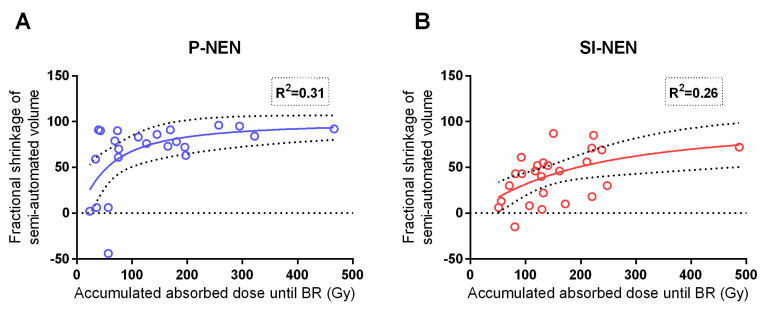
Comparison between tumor dose (Gy) and semi-automated volume (cm^3^) shrinkage in P-NENs versus SI-NENs. Accumulated absorbed dose (Gy) to the tumor versus the maximum fractional shrinkage of semi-automated volume (cm^3^) in P-NENs (**A**) (*n* = 23) and SI-NENs (**B**) (*n* = 25). P-NEN (*p* = 0.03), SI-NEN (*p* = 0.02). BR: best response.

**Table 1 cancers-13-00962-t001:** Characteristics for the 25 SI-NEN and 23 P-NEN study patients at baseline.

Variable		SI-NEN (*n* = 25)	P-NEN (*n* = 23)
Age	≤65 years	11	18
	>65 years	14	5
Krenning score	3	11	5
	4	14	13
	ND	0	5
Ki-67 index	≤2%	14	4
	3–20%	10	18
	>20%	1	1
	ND	0	0
Liver metastases (%)*	I (<10%)	2	3
	II (10–50%)	18	15
	III (>50%)	5	2
	IV (>50 + enlarged liver)	0	0
	ND		3
Bone metastases	No	11	15
	Yes	14	5
	ND	0	3
Extra-hepatic, non-skeletal metastases	No	3	8
	Yes	22	13
	ND	0	3
Hormonal symptoms	None	1	23
	One	9	
	Two	15	

* % of liver volume, according to therapy SPECT at first PRRT cycle.

**Table 2 cancers-13-00962-t002:** Kaplan–Meier analysis of progression-free survival between P-NENs and SI-NENs.

Group	Median Time (Months)	Lower 95%	Upper 95%	25% Failures	75% Failures
P-NEN patients	28.5	17	36	14.5	43.5
SI-NEN patients	30	26	44	23	48
Combined	29	27	36	20	46

**Table 3 cancers-13-00962-t003:** Fractional shrinkage (∆%) in P-NENs and SI-NENs from baseline (BL) to best response (BR) regarding tumor diameter, semi-automated tumor volume and RECIST 1.1 values.

		P-NEN Shrinkage (∆%) from BL to BR	SI-NEN Shrinkage (∆%) from BL to BR	95% Confidence Interval of the Mean Differences
Diameter	Median	−36	−15	
	Quartile range	−53 to −20	−21 to −3	
	Mean ± SEM	−37 ± 5	−15 ± 3	−33 to −11
Semi-automated volume	Median	−78	−44	
	Quartile range	−90 to −62	−56 to −18	
	Mean ± SEM	−65 ± 8	−40 ± 5	−41 to −10
RECIST 1.1	Median	−32	−13	
	Quartile range	−47 to −15	−19 to −4	
	Mean ± SEM	−32 ± 4	−12 ± 2	−27 to 13

**Table 4 cancers-13-00962-t004:** Decrease in the absorbed tumor dose (Gy) and in the ratio between absorbed tumor dose and administered activity of ^177^Lu-DOTATATE (Gy/GBq) from cycle 1 to cycle 3/4 (full seven-day dosimetry), in P-NEN and SI-NEN.

	Decrease in Absorbed Dose (Gy) from Cycle 1 to 3/4	CI 95%;	Decrease in Absorbed Dose Per Administered Activity (Gy/GBq) from Cycle 1 to 3/4	CI 95%;
P-NENs (mean ± SEM) *n* = 20	25.1 ± 5.5	34.5–15.6	3.0 ± 0.71	4.2–1.7
SI-NENs (mean ± SEM) *n* = 19	7.1 ± 4.5	11.4–2.8	0.9 ± 0.32	1.4–0.34

**Table 5 cancers-13-00962-t005:** Accumulated absorbed dose (Gy) from baseline to best response in regard to diameter, semi-automated (SA) volume and RECIST 1.1.

Tumor	Diameter (mm)	SA-Volume (mm^3^)	RECIST 1.1
P-NENs (mean ± SEM)	136.4 ± 23.6	139.6 ± 23.3	130.3 ± 20.4
SI-NENs (mean ± SEM)	167.5 ± 19	155.2 ± 18.5	152.3 ± 18.2

## Data Availability

Data are available on request to the corresponding author.

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
