# Peer review of "Peptide Receptor Radionuclide Therapy (PRRT) with ^177^Lu-DOTATATE; Differences in Tumor Dosimetry, Vascularity and Lesion Metrics in Pancreatic and Small Intestinal Neuroendocrine Neoplasms"

_cancers, 2021, doi:10.3390/cancers13050962_

Round 1

Reviewer 1 Report

Other article have been published using similar data. Publishing of clinical protocols in Cancers needs a specific clinical approach, in case collaborating with an experienced clinician. In this case it is important to clarify the possible predictive and/or prognostic value of the studied parameters.

Dosimetry, shrinkage and vascularization should related to grading, PFS, OS, clinical response, tumour burden, intensity of analogue uptake and glucose consumption. If not, we do not demonstrate their predictive or prognostic value. Without these parameters, data may be pure statistics.

Except for neo-adjuvant therapy, which is the interest to have tumour shrinkage at a certain dosage if it does not improve PFS and/or OS? In this case we also know that high-grade tumours can have early shrinkage but frequently will rapidly progress.

Is there a correlation between cumulative tumour dosage and outcome in different tumour grading?

Is the dose drop during cycles due to shrinkage, different receptor density or residence time reduction?

The association of tumour and critical organs dosimetry, together with the active tumour dose, may give the possibility to tailor therapies in order to allow a possible PRRT re-treatment!

I think you certainly have a treasure of interesting data in a five years follow up! Otherwise, once corrected some errors, the paper could be suitable for a niche physics or imaging journal.

I generally prefer the standard exposition: introduction, materials and methods, results, discussion, conclusions. This makes it easier for the readers to understand your work and arguments. Despite this, I don’t think it's mandatory

Since there are also G3 tumours, we are talking about NEN

Page 2, 2.1 patients: I do not understand why 4 patients who were stably in SD were removed from the PFS statistic.

Figure 1: More than PFS it seems to me the OS table

Conclusions: in the first sentence the verb is missing

Author Response

Comments and Suggestions for Authors

  1. Other articles have been published using similar data. Publishing of clinical protocols in Cancers needs a specific clinical approach, in case collaborating with an experienced clinician. In this case it is important to clarify the possible predictive and/or prognostic value of the studied parameters.

Reply: Yes, we have previously published several papers on kidney and bone marrow dosimetry (Sandstrom M et al. and Garske U et al.) a paper on dosimetry-tailored PRRT in 200 patients (Garske-Román U et al. 2018) and two papers on tumor dosimetry during PRRT (Ilan E et al. and Jahn U et al.). These studies have been either retrospective or prospective randomized but observational and not randomized.

  1. Dosimetry, shrinkage and vascularization should be related to grading, PFS, OS, clinical response, tumour burden, intensity of analogue uptake and glucose consumption. If not, we do not demonstrate their predictive or prognostic value. Without these parameters, data may be pure statistics.

Reply: Yes, a randomized controlled trial with PFS and OS as outcome parameters is of course the optimal basis in order to draw firm conclusions on the value of dosimetry versus a standardized PRRT protocol. In the “PRRT community” the research has however not come that far yet in the assessments of dosimetry (as mentioned in point 1 above), and currently these studies are merely observational to gather data and try to understand the different mechanisms for the effects of PRRT and to identify the various factors that may have an impact on the patients´ outcome. The registered and marketed preparation (Lutathera) with its prescribed 4 x 7,4 GBq PRRT protocol is where we are today, however, this is not a treatment regime which is optimal for all NET types and all patients, and factors such as tumour load, amount of activity per PRRT cycle (and variations in administered activity between cycles), interval between cycles and amount of administered peptide and dosimetry monitoring, are all factors that need to be assessed in order to adapt PRRT-protocols to better suit the individual patient. We have tried to reflect on this in the introduction and discussion sections (Lines 78-88, Lines 241-264).

  1. Except for neo-adjuvant therapy, which is the interest to have tumour shrinkage at a certain dosage if it does not improve PFS and/or OS? In this case we also know that high-grade tumours can have early shrinkage but frequently will rapidly progress.

Reply: There are so far merely a limited number of publications on tumour dosimetry during PRRT and yet no randomized controlled comparative trial has been performed on dosimetry-guided PRRT versus a standardized PRRT protocol. There are therefore no data to support the value of dosimetry-tailored PRRT and this is what our group is investigating, so far merely in observational studies. As the reviewer pertinently points out in the example of patients with high-grade tumours who initial respond well on therapy but than very rapidly progress, no group, neither our or any other group, have this far conducted a randomized study showing that patients who receive as many PRRT cycles as possible, tailored by dosimetry, will have better outcome (PFS/OS) than those who receive the standard protocol of 4 x 7,4 GBq 177Lu-DOTATATE. However, the results from our prospective but non-randomized PRRT study in 200 patients strongly indicate that this is the case (Garske-Román et al. EJNMMI 2018), but definitely needs to be further studied. This includes also tumour dosimetry, the focus of the present manuscript.

  1. Is there a correlation between cumulative tumour dosage and outcome in different tumour grading?

Reply: Unfortunately, the number of patients was too small to permit sub-analyses based on their tumour grade (G1 versus G2 versus G3 etc.).

  1. Is the dose drop during cycles due to shrinkage, different receptor density or residence time reduction?

Reply: This we believe is a very interesting observation, the reasons for which are not clear. However, as we mention in the discussion section, changes during PRRT in tumor vascularization, for example because of varying vessel density may affect not only the radioactivity delivery and distributed throughout the tumor, but also the tumor oxygenation, which may impact the effects of the delivered radiation. Please see the Discussion section (Lines 254-257). The only way to study this question further would be a prospective study involving biopsies after each cycle. This would be very interesting, but hard to pursue in a clinical setting.

  1. The association of tumour and critical organs dosimetry, together with the active tumour dose, may give the possibility to tailor therapies in order to allow a possible PRRT re-treatment!

Reply: Yes, we agree. This is what we are trying to achieve by firstly exploring our existing data on normal organ and tumour dosimetry and, from what we learn from these together with published data from other groups, we intend to design an RCT to be able to perform a thorough comparative assessment of PRRT protocols, which also includes the re-treatment situation.

  1. I think you certainly have a treasure of interesting data in a five-years follow up! Otherwise, once corrected some errors, the paper could be suitable for a niche physics or imaging journal.

Reply: Thanks! Yes, our data have been collected mainly in connection with the above-mentioned prospective dosimetry tailored PRRT study, but in fact all our patients undergo normal tissue dosimetry, and we consider it valuable to also explore and compile our data retrospectively, and as the reviewer points out, we have a long follow-up time for most of our patients. We have for the current manuscript discussed how to best reach the readers who are involved in PRRT and may benefit from sharing our data. We concluded that a clinical journal would probably be the better alternative, and we chose Cancers as a highly regarded journal for which the scope of our paper most probably would be suitable to reach the treating physicians who prescribe PRRT. We think that an imaging journal or a physics journal therefore may not be as good.

  1. I generally prefer the standard exposition: introduction, materials and methods, results, discussion, conclusions. This makes it easier for the readers to understand your work and arguments. Despite this, I don’t think it's mandatory

Reply: We have strictly followed the guidelines of the Cancers journal which state that the manuscript sections should have this order, with the Materials and Methods section inserted after the Discussion.

  1. Since there are also G3 tumours, we are talking about NEN

Reply: Yes, this is also a very pertinent comment. We have now updated the nomenclature to NEN throughout the manuscript.

  1. Page 2, 2.1 patients: I do not understand why 4 patients who were stably in SD were removed from the PFS statistic.

Reply: Thank you for pointing this out. We have therefore rephrased the sentence regarding these results and hope that this is now clearer (Lines 86-88).

  1. Figure 1: More than PFS it seems to me the OS table

Reply: We are sorry but we fail to understand this comment since we cannot find that we have reported any results on OS. However, following suggestion from another reviewer we decided to exclude Figure 1 and instead report results on PFS and follow-up time only in the text.

  1. Conclusions: in the first sentence the verb is missing

Reply: The verb in the first sentence of the Conclusion section should be “differed”. Please see (Line 403).

Reviewer 2 Report

The manuscript by Jahn and colleagues describes the impact of dosimetry on NET tumor shrinkage. Authors evaluate a cohort comprising 25 patients with SI-NET and 23 patients with pan-NET who underwent dosimetry-guided PRRT with 177Lu-DOTATATE in the context of the EudraCT 2009-012260-14 study. First, Authors assessed the correlation between accumulated administered activity and tumor shrinkage. Second, they explored the impact of absorbed dose (in 1 selected tumor lesion) on tumor shrinkage. Third, they investigated the effects of PRRT on tumor vascularization by using the HU tumor-to-aorta ratio. Overall, the manuscript is well written and informative, focusing on an intriguing concept (the role of dosimetry in tailoring the treatment with 177Lu-DOTATATE). Tables and figures are informative (but see specific comments below). Conclusions are balanced and based on the evidence provided in the results section. References are appropriate and updated.

Specific comments:

  1. There are a few typos throughout the manuscript (i.e., line 22, line 167, line 198, etc) that should be corrected.
  2. Line 60. NETTER-1
  3. Line 84. I guess Authors present here the range of the follow-up. This should be specified. Please also report the median duration of the follow-up.
  4. It is not clear to me why Authors chose TTP (instead of PFS) as survival endpoint. Moreover, the definition of TTP in the M&M is scant, and does not allow to understand how were events exactly defined. Please, refer to FDA definitions as a guidance.
  5.  Figure 1. TTP and PFS are different - the x axis here does not have any sense. Moreover, I am not sure to understand what is the goal of this figure. Why Authors deem so important to show that there is no difference in terms of TTP (PFS?) between SI-NET and panNET patients? Perhaps this figure could be trimmed (or can be shown as supplementary material). Not sure about the meaning of 25%-75% failures in the table below the KM.
  6. All abbreviations should be defined at their first appearance throughout the text (i.e., SEM - among others)
  7. Figure 2. Why did Authors deem this figure crucial in their report? I am not sure that this figure is going to make a real difference within this manuscript. It can be presented as supplementary material - or just trimmed.
  8. Please avoid redundancy between M&M and results (i.e. when dealing with different methods of tumor volume calculation).
  9. Base line should read baseline (line 104 and elsewhere in the manuscript)
  10. Authors must always include the calculated p-value when reporting with statistical analyses (i.e., lines 106-107). Please report the exact p-values when differences are significant, use p>0.05 when the statistical threshold is not met.
  11. Once defined at their first appearance, abbreviations (i.e., delta%) can be used without the need of further definitions in both the text, tables and figures 
  12. Table 1. Please correct RESIST in RECIST.
  13. I would suggest including Table 4 in the context of the section "Patients" of the Results - readers will otherwise struggle to understand what are the clinical-pathological characteristics of the studied cohort until the end of the manuscript.
  14. Line 121. Is the difference statistically significant? Again, please always report p-values.
  15. Figure 3. The is a typo in the y and x axes.
  16. Line 130-131. Not sure about the denominators here (21 and 20). Why? Moreover, did the vascularization show a similar behavior when the radius volume/RECIST were analyzed?
  17. Figure 4A. I can count only 22 circles. How many patients were analyzed here?
  18. Figure 5. Not sure about the numbers here. In the text Authors state that 19 patients had repeat dosimetry measurements. In the figure legend they state that 20 panNET patients were analyzed. I can count 21 dots at cycle 1 and 19 at subsequent cycles in the figure. At this regard, Authors should include only patients with repeat evaluations. The Wilcoxon rank sum test should be used here for statistical analysis.
  19. Table 2. If I am interpreting the data in the right way, there is an increase in the absorbed dose per administered activity from cycle 1 to cycles 3/4 in the SI-NET cohort. Do Authors have an explanation for this phenomenon?
  20. Figure 6. There is a typo in the y axis. There is no bar for cycle 7 in the panNET cohort. If this timing is not relevant for panNETs, it can be erased.
  21. Figure 7. Apparently, this figure focuses on patients with low to moderate tumor burden. There is no definition in the text for such a group. Moreover, I am not sure about the numbers here - I cannot count 20 lines in the spider plot referring to panNET for example.
  22. Lines 193-194. There is a discrepancy between what Authors state here and Table 3, where the absorbed dose appear to be higher in SI-NETs as compared with panNETs.
  23. Line 209. Based on the p-values provided in the figure legend, the correlation appears to be significant (not close to significant)
  24. Lines 266-271. Please rephrase.
  25. Line 280. RECIST criteria do not provide information on the tumor burden.
  26. Can you identify a threshold of absorbed dose capable of predicting PR? and maybe longer PFS?
  27. Table 4, line 329. There is some confusion regarding the numbers. Authors enrolled 23 patients with panNETs
  28. Line 337. Please report the number of patients who received concomitant somatostatin analogs.
  29. Line 347. VOIs needs to be defined.
  30. Lines 354-355. Not sure about the mention to CTCAE criteria in a manuscript that is not dealing with toxicities.
  31. Line 362. Did all patients receive both CT and MRI scans? Is there any correlation between DWI coefficient and tumor attenuation?
  32. Line 396. Not sure what is Graf Pad
  33. Discussion. Intriguingly, Authors report a decrease of tumor vascularization throughout the treatment with 177Lu-DOTATATE. However, Authors are well aware of the NET vascularization paradox (well differentiated tumors are more vascularized than poorly differentiated ones). Many reports have described a progressive increase of the Ki-67 index over treatment. A few studies have demonstrated a progressive accumulation of mutations within NET cells during treatment with PRRT/chemotherapy. Any thoughts?

Author Response

Reviewer 2

Comments and Suggestions for Authors

The manuscript by Jahn and colleagues describes the impact of dosimetry on NET tumor shrinkage. Authors evaluate a cohort comprising 25 patients with SI-NET and 23 patients with pan-NET who underwent dosimetry-guided PRRT with 177Lu-DOTATATE in the context of the EudraCT 2009-012260-14 study. First, Authors assessed the correlation between accumulated administered activity and tumor shrinkage. Second, they explored the impact of absorbed dose (in 1 selected tumor lesion) on tumor shrinkage. Third, they investigated the effects of PRRT on tumor vascularization by using the HU tumor-to-aorta ratio. Overall, the manuscript is well written and informative, focusing on an intriguing concept (the role of dosimetry in tailoring the treatment with 177Lu-DOTATATE). Tables and figures are informative (but see specific comments below). Conclusions are balanced and based on the evidence provided in the results section. References are appropriate and updated.

Reply: We would first sincerely like to thank the reviewer for his/her thorough, professional and excellent review!!! We are very thankful for the reviewer´s work and all her/his very pertinent comments, which have been very helpful and we believe that because of this our revised manuscript has been much improved.

Specific comments:

  1. There are a few typos throughout the manuscript (i.e., line 22, line 167, line 198, etc) that should be corrected.

Reply: We have once again proofread the text and hope that we have found and corrected all typos.

  1. Line 60. NETTER-1

Reply: This has now been corrected (Line 46, Line 61)

  1. Line 84. I guess Authors present here the range of the follow-up. This should be specified. Please also report the median duration of the follow-up.

Reply: This has now been corrected (Line 86, Lines 86-88)

  1. It is not clear to me why Authors chose TTP (instead of PFS) as survival endpoint. Moreover, the definition of TTP in the M&M is scant, and does not allow to understand how were events exactly defined. Please, refer to FDA definitions as a guidance.

Reply: To avoid confusion we have now only reported results on PFS. Figure 1 has also after further consideration been found redundant. The definition is restated in section 4.6. (Lines 393-394).

  1.  Figure 1. TTP and PFS are different - the x axis here does not have any sense. Moreover, I am not sure to understand what is the goal of this figure. Why Authors deem so important to show that there is no difference in terms of TTP (PFS?) between SI-NET and panNET patients? Perhaps this figure could be trimmed (or can be shown as supplementary material). Not sure about the meaning of 25%-75% failures in the table below the KM.

Reply: To avoid confusion we have now only reported results on PFS. Figure 1 has also after further consideration been found redundant, thus we believe that to include information about failures would be of value (Lines 91-92).

  1. All abbreviations should be defined at their first appearance throughout the text (i.e., SEM - among others)

Reply: This has now been adjusted throughout the manuscript.

  1. Figure 2. Why did Authors deem this figure crucial in their report? I am not sure that this figure is going to make a real difference within this manuscript. It can be presented as supplementary material - or just trimmed.

Reply: No, this is not crucial information and the figure has now been deleted. However, we comment in the text on the fact that the deviation of the data became much less pronounced with measurements of the tumor volume rather than diameter.

  1. Please avoid redundancy between M&M and results (i.e. when dealing with different methods of tumor volume calculation).

Reply: This has now been adjusted throughout the manuscript. However, because of the journal´s lay-out with the Results section inserted before the Materials and Methods section, we have in order to facilitate for the reader, considered it necessary to relate, although very shortly, occasional methodological details also in the results section, such as the semiautomated tumor volume measurements versus those based on the transverse diameter, the tumor heterogeneity based on native CT attenuation and tumor vascularization based on the arterial CT tumor-to-aorta ratio (Lines 94-96 Lines 122-125). We hope that this is acceptable.

  1. Base line should read baseline (line 104 and elsewhere in the manuscript)

Reply: This has now been adjusted throughout the manuscript.

  1. Authors must always include the calculated p-value when reporting with statistical analyses (i.e., lines 106-107). Please report the exact p-values when differences are significant, use p>0.05 when the statistical threshold is not met.

Reply: This has now been adjusted throughout the manuscript.

  1. Once defined at their first appearance, abbreviations (i.e., delta%) can be used without the need of further definitions in both the text, tables and figures 

Reply: This has now been adjusted throughout the manuscript.

  1. Table 1. Please correct RESIST in RECIST.

Reply: This has now been adjusted (Table 1 is now Table 3) (Line 110).

  1. I would suggest including Table 4 in the context of the section "Patients" of the Results - readers will otherwise struggle to understand what are the clinical-pathological characteristics of the studied cohort until the end of the manuscript.

Reply: Thanks! Very good suggestion. This has now been adjusted (Table 4 is now Table 1).

  1. Line 121. Is the difference statistically significant? Again, please always report p-values.

Reply: Yes, this is significant (p<0.001) and has now been reported (Line 119).

  1. Figure 3. The is a typo in the y and x axes.

Reply: This has now been adjusted (Figure 3 is now Figure 1) (Lines 84-85).

  1. Line 130-131. Not sure about the denominators here (21 and 20). Why? Moreover, did the vascularization show a similar behavior when the radius volume/RECIST were analyzed?

Reply: Thank you for pointing out the necessity of clarifying the number of tumors assessed for vascularization. There were 2 P-NEN patients and 5 SI-NEN patients who did not receive i.v. contrast at CT examination at the time of best response. Consequently, there were 21 P-NEN patients and 20 SI-NEN patients included in the analysis of tumor vascularization. This has now been clarified in the text (Lines 127-128).

Regarding the second part of the question, the time interval over which the changes in tumor vascularization was measured was from baseline to best response, as regards the semi-automated tumor volume. Since changes in tumor volume in our work was found most reliable as a measure of response we did therefore not assess changes in vascularization between baseline and best response regarding tumor radius/volume/RECIST 1.1.

  1. Figure 4A. I can count only 22 circles. How many patients were analyzed here?

Reply: Two patient from each NEN-group have identical values, and can therefore not be seen as separate circles (because of overlay). The figure is based on 23 P-NENs and 25 SI-NENs. This has now been pointed out in the figure legend. (Figure 4 A is now Figure 2A.)

  1. Figure 5. Not sure about the numbers here. In the text Authors state that 19 patients had repeat dosimetry measurements. In the figure legend they state that 20 panNET patients were analyzed. I can count 21 dots at cycle 1 and 19 at subsequent cycles in the figure. At this regard, Authors should include only patients with repeat evaluations. The Wilcoxon rank sum test should be used here for statistical analysis.

Reply: Quite correctly there are 20 P-NENs and 19 SI-NENs in this figure, as stated in the figure legend, and the text in the results section has now been corrected. The extra dots in both figures were explained by the highlighting of the outliers, which for unknown reasons with the use of the JMP software resulted in duplication of these dots. To avoid this confusion, related to the software, we decided to modify Figure 5 and present only a boxplot. (Figure 5 is now Figure 3)

The statistical testing was performed using Wilcoxon rank sum test and showed a significant decline in P-NENs (p<0.01) and non-significance in SI-NENs (p>0.05) (Line 171).

  1. Table 2. If I am interpreting the data in the right way, there is an increase in the absorbed dose per administered activity from cycle 1 to cycles 3/4 in the SI-NET cohort. Do Authors have an explanation for this phenomenon?

Reply: Table 2 has by request now been renumbered to Table 4. The table actually shows a decrease in absorbed tumor dose (Gy) and of absorbed dose per administered activity (Gy/GBq) over the course of PRRT (at cycles 3/4 as compared to cycle 1). This we believe is a very interesting observation, the reasons for which are not clear. However, as we mention in the discussion section, changes during PRRT in tumour vascularization, for example because of vessel density, may affect not only the radioactivity delivery and distribution throughout the tumor but also the tumour oxygenation, which may impact the effects of the delivered radiation.

  1. Figure 6. There is a typo in the y axis. There is no bar for cycle 7 in the panNET cohort. If this timing is not relevant for panNETs, it can be erased.

Reply: This has now been corrected (Figure 6 is now Figure 4). Unfortunately, the JMP software cannot be adjusted to erase letters or numbers in only one part of the combined graph.

  1. Figure 7. Apparently, this figure focuses on patients with low to moderate tumor burden. There is no definition in the text for such a group. Moreover, I am not sure about the numbers here - I cannot count 20 lines in the spider plot referring to panNET for example.

Reply: Thanks for pointing this out! (Figure 7 now Figure 5) The tumour burden was assessed on scintigraphy during the first cycle of PRRT as noted in Table 1. We have now added in the materials and methods section how the patient’s tumor burden was graded based on therapy scans during cycle 1 (scintigraphy during PRRT). A short comment on this has also been added to the figure legend in Figure 5. Please see the last sentence in section 4.1. (Lines 329-332).

  1. Lines 193-194. There is a discrepancy between what Authors state here and Table 3, where the absorbed dose appears to be higher in SI-NETs as compared with panNETs.

Reply: Thanks! This is a very pertinent comment! Unfortunately, there was some confusion with these numbers. All these numbers in the Table have now been recalculated and double-checked and have been corrected accordingly. (Table 3 is now Table 5) (Lines 200-202).

  1. Line 209. Based on the p-values provided in the figure legend, the correlation appears to be significant (not close to significant)

Reply: Thanks! We completely agree and this has been corrected (Line 214).

  1. Lines 266-271. Please rephrase.

Reply: Yes, we agree that this was not very well expressed and has now been rephrased (Lines 260-264).

  1. Line 280. RECIST criteria do not provide information on the tumor burden.

Reply: No, this is absolutely correct and this is not what we intend to express and this sentence needs to be rephrased. What we mean to express is that as opposed to data on one single tumor lesion, we also wanted to present data which considers more of the patients´ tumor disease, although not the total tumor burden. To achieve this, we applied RECIST 1.1, which at least should include measurements of the two largest tumors in each organ and a total of 5 tumors. We have now rephrased this sentence (Line 281).

  1. Can you identify a threshold of absorbed dose capable of predicting PR? and maybe longer PFS?

Reply: No, our results do not allow for this. We need to accumulate more data. There is though, a conceivable range of response patterns. We should mention in this respect, that we have occasional clinical observations of (near-to) complete regression of tumor uptake after only one therapy cycle, both in a patient with a G3 pancreatic NEN and a G1 SI NET. None of the patients were examined according to the dosimetry protocol described in this paper.

  1. Table 4, line 329. There is some confusion regarding the numbers. Authors enrolled 23 patients with panNETs.

Reply: Thank you. This typo has now been corrected, (Table 4 now Table 1) (Line 84).

  1. Line 337. Please report the number of patients who received concomitant somatostatin analogs.

Reply: All patients were on somatostatin analogs and we have rephrased this sentence accordingly (Lines 338-339).

  1. Line 347. VOIs needs to be defined.

Reply: “Volume of interest” has now been written out instead of the abbreviation (VOI) (Line 349).

  1. Lines 354-355. Not sure about the mention to CTCAE criteria in a manuscript that is not dealing with toxicities.

Reply: Yes, this is correct. Redundant information. This sentence has now been rephrased and does no longer mention the CTCAE criteria (Lines 356-357).

  1. Line 362. Did all patients receive both CT and MRI scans? Is there any correlation between DWI coefficient and tumor attenuation?

Reply: Only one patient was monitored using MRI and therefore no evaluation of DWI could be performed to measure the ADC values or changes of ADC over the course of PRRT.

  1. Line 396. Not sure what is Graf Pad

Reply: Graf Pad is “Graf Pad Prism” a computer software to compute statistics and make graphs. This has now been clarified in the text (Line 400).

  1. Discussion. Intriguingly, Authors report a decrease of tumor vascularization throughout the treatment with 177Lu-DOTATATE. However, Authors are well aware of the NET vascularization paradox (well differentiated tumors are more vascularized than poorly differentiated ones). Many reports have described a progressive increase of the Ki-67 index over treatment. A few studies have demonstrated a progressive accumulation of mutations within NET cells during treatment with PRRT/chemotherapy. Any thoughts?

Reply: Yes, we agree, both research and clinical work within the field of NENs is indeed intriguing! Although PRRT has been available for quite some time, and we have unquestionable come a long way, there are still many factors which most probably impacts the treatment and the patient outcome and which we need to identify and understand and learn from how to best manage in order to optimize the PRRT protocols. The optimal protocol is probably different depending on the patients NEN type and probably also on the tumour load. There are several obvious parameters that should be evaluated such as the amount of administered activity per cycle, similar or different amounts of activity per cycle, the interval between cycles, the amount of administered peptide per cycle, fixed number of cycles, maximum number of cycles according to normal tissue and tumour dosimetry, etc. These parameters will most probably be studied over the next couple of years and the PRRT protocols modified accordingly. However, the most intriguing factors are the intrinsic tumour related ones, such as proliferation (and, yes, we have observed that it is not uncommon with an increase in Ki-67 over the course of treatment, probably more frequent in P-NETs), degree of vascularization versus quiescent cells and necrosis, the relation between different tumour cells and non-tumour cells within the tumour, genetics and epigenetics etc. These factors are much more challenging to assess but are equally important. In order not to make the discussion too long and also, to avoid the risk of crossing the border to speculation, we have refrained from discussing factors that we consider not to be directly related to our findings.

Your interesting comment on progressive accumulation of mutations within NET cells during treatment calls for further reflection. A notorious problem in oncology is remaining sub-clinical disease. The research in our group has this far indicated that oncologists need to adapt their mindset because PRRT combines the elements of systemic treatment and radiation. The knowledge gathered from external beam radiation is not readily applicable to PRRT (over-treatment will increase local, but not necessarily systemic toxicity), nor the knowledge of systemic toxicity related to chemotherapy. In the setting of PRRT, decrease of tumour volume/ uptake potentially increases radiation to healthy tissues, especially important for radiosensitive organs such as the bone marrow. Repetitive cycles of chemotherapy as well as external beam radiotherapy are administered in order to minimize/ extinct residual disease and by this, the risk for remaining tumor cell clones with potentially worse characteristics, such as increased proliferation, increases. For internal radiotherapy such as PRRT, we have to find new ways of defining the optimal cut-off between reduction of tumour volume and toxicity. This means not only defining the upper tolerance level of dose to sensitive organs, or to find optimal absorbed tumor dose levels. This means also learning to use imaging during therapy in order to be able to discontinue therapy in order to prevent toxicity. This was the reason to settle for tumour regression by more than 90% of the original tumour burden (on post-therapy imaging) as the criterion to discontinue PRRT in our prospective study (Garske-Román et al. 2018).

Round 2

Reviewer 1 Report

In my opinion the work suffers from numerous publications of partial data and lacks originality as well as clinical insight. I think, the article is more suitable for publication in a niche journal.

Reviewer 2 Report

Authors have addressed all my criticisms. The manuscript has been improved.